# Analysis of the Emotional Identification Mechanism of Campus Edible Landscape from the Perspective of Emotional Geography: An Empirical Study of a Chinese University Town

**DOI:** 10.3390/ijerph191811425

**Published:** 2022-09-10

**Authors:** Jinping Lin, Meiqi Zhou, Huasong Luo, Bowen Zhang, Jiajia Feng, Qi Yi

**Affiliations:** 1School of Earth Sciences, Yunnan University, Kunming 650091, China; 2Faculty of Geography, Yunnan Normal University, Kunming 650500, China

**Keywords:** edible campus, emotional geography, SEM, campus landscape, Chinese university town

## Abstract

Against the background of “the emotional turn” in geography, the study of emotional identification is attracting increasing attention among researchers. Edible landscape resources can satisfy the emotional needs of teachers and students by enabling them to experience pastoral landscapes that carry cultural and landscape values to campus environments. Based on a questionnaire survey of 419 students and teachers at Chenggong University Town in China, this study improved the structural equation modeling (SEM) method to construct a model to analyze the emotional identification mechanism of the campus edible landscape. The research found that emotional identification played an intermediary role between perception and behavioral intention, manifested as an association mechanism in which surface values influence perception, perception influences emotional identification, and emotional identification influences behavioral intention. The emotional identification model revealed the relationship between teachers and students’ emotional identification and the value of campus edible landscape resources for the first time. It also uncovered the universality of the association mechanism in the research of emotional geography.

## 1. Introduction

Landscapes and food are both vital and necessary topics [1]. The edible landscape is a multifunctional landscaping method that has emerged recently. As a landscape composed of plant species that can be eaten by humans, it emphasizes the function and value of the landscape and is characterized by edibility, ornamentation, and economy [2,3]. In the 19th century, the trend of modern pastoral cities emerged in landscape design and urban planning [2]. In the context of the conflict between urban landscape construction and agricultural production, the edible landscape takes into account the functions of both, and it meets the need for coordination between healthy and ecological urban agriculture and the living environment.

However, not all edible landscapes are justified. For example, the food safety of some edible landscapes planted next to roads may be questionable. The rational development of edible landscapes needs to meet the following principles. First, it needs to meet the principles of environmental friendliness and organic cultivation. The essence of an edible landscape is that it is ecological and sustainable, and its planning and planting must conform to the principles of environmental protection. In particular, edible landscapes on campuses should be cultivated organically to ensure food safety. Second, the diversified functions and values of edible landscapes should be fully developed, and multi-functional synergies should be realized in terms of ornamental and practical uses as well as popular science and education.

The United States is one of the world’s most advanced countries in the development and utilization of edible landscapes [4,5,6]. Relying on a well-organized urban agricultural system, it has developed representative edible landscape models such as rooftop farms and vertical farms [7]. British community orchards, German life gardens, French ecological farms, Japanese urban farms, and Dutch edible mobile huts are also examples of the rational development of edible landscapes in other countries [7]. In the 1990s, the ecological campus project was initiated in China. The edible campus landscape represents an important breakthrough in combining agriculture, landscape, and education, which, through entertainment, contributes to popular science education and scientific research [8,9,10].

Since teaching activities are held on campus from Monday to Friday, teachers and students have more opportunities to interact with the campus landscape, especially edible landscapes. Campus edible landscapes integrate viewing, eating, and enjoyment, and they provide teachers and students with an opportunity for human–land interaction. Interacting with campus edible landscapes can meet students and teachers’ emotional needs for sightseeing, work, and rest, and it can help with mood adjustment. In this context, universities in various university towns have promoted the construction of campus edible landscapes to enhance the visibility of their campus tourism brands.

Teachers and students’ emotional identification is defined as the emotional experience of whether the objective reality meets their needs. On the one hand, college students have typically relocated from home [11,12]. The changing emotional characteristics of college students are important for analyzing the dynamics of their mental health, especially their identification with the university and the place to which they moved [13]. On the other hand, the career decision-making of college students is closely related to emotional identification [14,15]. Teachers and students at universities first form a formal identification by visiting campus edible landscapes. If there is a similarity between attitudes and perceptions of edible landscapes and related food, an idealized identification is generated. Formal and idealized identification affect behavioral intentions, showing the material identification of buying or enjoying relevant food and protecting edible landscapes [16].

This study explores the function and value of campus edible landscape resources from the perspective of teachers and students’ emotional identification. An improved structural equation modeling (SEM) method (improved in terms of case-site selection and model validation) is used to construct an emotional identification model. The empirical research is carried out in Chenggong University Town. Using the data, we revise and verify the emotional identification model and analyze the influence mechanism of the perception, emotional identification, and behavioral intention of teachers and students. 

This study represents a novel approach to the study of emotional geography. The findings offer insights on enhancing campus tourism brands and promoting the construction of ecological campuses. Moreover, this research expands researchers’ horizons and ideas and advances the deepening, systematization, and theorization of subsequent emotional geography research.

## 2. Literature Review

### 2.1. Campus Edible Landscapes

The planning and construction of edible landscapes promote the scientific and rational allocation of landscape resources in universities and enhance the value of campus landscapes in multiple ways, including aesthetic appeal, scientific research, ecology, and economy. A campus edible landscape refers to a landscape composed of plant species grown on campus that can be eaten by humans. Campus edible landscapes often have a large scale; low-density and scattered edible plants would be categorized as an ornamental rather than a cultivated edible landscape. The concept of a campus edible landscape originated in the West, resulting in a closer connection between the campus and outside world [17,18,19,20]. The construction of campus landscapes has changed the traditional rigidity of campus landscapes to landscapes characterized by openness, enabling better landscape gradation on campuses. Western university campuses have developed over the centuries. Accordingly, campus edible landscape construction has matured, with universities such as Michigan State University becoming models. With the constant collision of Chinese traditional culture and Western campus planning concepts, Chinese universities have transformed from ancient academies to university towns with clear functional divisions. The land area of universities continues to expand. Thus, determining how to allocate campus landscape resources scientifically and rationally has become an important research topic at modern universities. The research on campus edible landscapes in Western countries precedes that in China, which has a fresher experience in their construction and utilization. Research on campus edible landscapes is limited in China, focusing on qualitative description and analysis; the research content is also less extensive. Most studies have been limited to the planning and design of campus edible landscapes [21,22,23]. Research on the emotional interaction between the campus landscape and teachers and students is rare. With the exception of Yang et al. [24], few scholars have made a preliminary examination of the emotional communication between campus edible landscapes and people from the perspective of perceptual interaction.

Different from the edible landscapes in cities and suburbs, campus edible landscapes are smaller in area, and more emphasis is placed on eliminating additives, ensuring food safety, and promoting public welfare, scientific research, and education. These landscapes mainly serve teachers and students, so the function of edible landscapes on campus is richer. Drawing on domestic and foreign studies on tourism resources evaluation [25], the value of campus edible landscape resources was condensed accordingly.

Campus edible landscapes have ornamental and recreational value. They comprise something tangible, colorful, and with body with perceptible vegetation and represent the characteristic combination of viewing and eating [26]. Campus edible landscapes have a recreational spatial form and social and cultural connotation, which trigger psychological reflections in terms of teachers and students’ tastes, imagination, and identification.

Campus edible landscapes have educational value and contribute to scientific research. They can be used as a training and research base. Moreover, they have a unique science education value, in that they exhibit characteristics of agricultural culture and landscape culture [27]. They not only enrich the form of teaching activities, but they also compensate for the lack of general agricultural knowledge among contemporary university students. They create an atmosphere of agricultural production, and they promote comprehensive education on morality, intelligence, physical fitness, aesthetics, and labor [28].

Campus edible landscapes have environmentally sustainable value. They have a good regional combination of resources and must consider the presence of pollution and public hazards to create a green, safe, and healthy campus landscape environment. Food safety takes precedence over everything else; thus, the university should follow the principle of ecology from pre-planning to post-care and uphold the sustainable development concept of ecological and environmental protection.

Campus edible landscapes have economic value. Edible plants are mostly cash crops, enriching the types of agricultural products available at the school and bringing economic benefits. Furthermore, the self-sufficiency model promotes the specialization of school food.

Campus edible landscapes have emotional identification value. The emotional identification of teachers and students is a virtuous closed *emotion-landscape* loop between their emotions and the campus edible landscape, that is, the virtuous cycle of the emotion-landscape relationship. The ornamental, experiential, and interactive features of campus edible landscapes provide teachers and students with the opportunity to interact with the scenery. Students and teachers emotionally accept, understand, and identify with the function and value of campus edible landscapes, generating positive environmental protection identification and service recognition, which manifest as love, attachment, and satisfaction, promoting positive behavioral intentions. Because of their unique combination of values, including appreciation, recreation, scientific research and education, ecology and sustainability, and economic value, campus edible landscapes can exert special emotional identification value.

### 2.2. Emotional Identification

The geographer Tuan created the important concept of place and believed that people’s emotions are closely related to place [29]. Emotional geography continues humanistic geography’s focus on place. Places contain a wealth of human emotions, such as topophilia, sense of place, and place attachment. In addition, emotional identification is also one of the important perspectives of emotional geography in place studies.

Emotional identification refers to the process of emotional sharing and convergence when objective things meet one’s own needs. Emotional identification is one of the main research fields of emotional geography, and it can be divided into material identification, idealized identification, and formal identification. Anderson and Smith introduced the concept of emotional geography in 2001, marking the beginning of emotional geography research [30]. Davidson et al.’s monograph, *Emotional Geography*, published in 2005, was a pioneering work that provided a detailed and systematic account of emotional geography [31]. The launch of *Emotional Space and Society* in 2008 opened a platform for the exploration of emotional spatiality and sociality, which successively reported systematic and comprehensive research results on emotional geography. Davidson and Smith and Parr et al. [32,33] proposed the study of emotional geography from the dimensions of space and society and space and time. Kearney divided emotional geography into emotional closeness and emotional alienation [34]. Subsequently, Kearney and Bradley proposed that emotional geography should aim to explore how human emotions affect cultural groups [35]. In China, Lin introduced the concept of emotional geography as the study of the interrelationships and important influences among people, emotions, and places [36]. Jian et al. noted that the main goal of the emotional geography field is to explore the relationships among people, emotions, and space [37]. Although scholars do not have a uniform understanding of the concept of emotional geography, there is a consensus on three dimensions: people, affect, and place (space) [38,39]. 

Hargreaves, a Canadian educator who focuses on teachers’ emotions [40], argues that human emotional interactions depend on spatial proximity or distance and proposes a theoretical framework of emotional geography, namely, political distance, professional distance, moral distance, physical distance, and sociocultural distance [41]. Teaching activity sites are dynamic, interactive, and reflective pedagogical spaces [42] which have received considerable scholarly attention in recent years. Pyndiah uses events such as the Holocaust as case studies to teach, stimulate student curiosity, and lead them to engage in discussions. The study reveals the paths of emotional interaction and emotional transmission between teachers and students in the teaching space [42]. 

Emotional geography investigates the interrelationship between human emotions and places and explores the patterns, causes, and trends of temporal and spatial changes in emotions. The relationship between people and campuses is an important research topic in modern human geography. Emotion is an essential part of the relationship between teachers and students. The influence mechanism between the campus landscape and teachers and students’ emotions is a new research direction. In general, the emotional identification mechanisms of teachers and students are less analyzed, and it is relevant to explore the emotional identification model in the context of campus edible landscapes.

### 2.3. Structural Equation Modeling Method

At present, scholars primarily apply the SEM method in the research fields of influence factors, evaluation index systems, satisfaction, and competitiveness, and there are fewer studies on the construction of SEMs for emotional identification. Most research on emotional identification has focused on the relationship between consumer emotional experience, patriotic emotions, and identity [43,44,45,46], and some scholars have used emotional identification as one of the constructive dimensions of local identity [47]. However, research on the emotional identification of campus edible landscapes is scarce. Social identity and place theories provide the theoretical basis for emotional identification studies [48,49]. However, determining how to correlate the emotional identification of teachers and students with the value of campus edible landscapes is a key scientific question to improve SEM methods.

## 3. Materials and Methods

### 3.1. Study Case

Edible landscapes are common in urban communities, courtyards, and other small areas. However, large-scale edible landscape development in colleges and universities is rare. At present, the homogeneity of landscape planning and design in Chinese colleges and universities restricts the personalized and diversified development of campus edible landscapes. In the 1930s, Tsinghua University planned and designed an edible landscape. Subsequently, Wuhan University, Shenyang Jianzhu University, Yunnan University, and others boldly implemented and innovated campus edible landscapes, forming an open campus tourism mode in China.

At the end of the 20th century, the number of students in Chinese universities grew rapidly. After the first university town was born in Langfang, more than seventy university towns were built. Chenggong University Town is located in the biodiversity-rich Yunnan Province of China, with seven undergraduate and two vocational universities and an enrollment of approximately 150,000 students. A comfortable climate and sufficient reserve land resources have promoted the development of edible campus landscapes. Chenggong University Town has become one of the richest edible campus landscapes in China. 

Located in Chenggong University Town, Yunnan University has been selected as one of the top 10 most beautiful universities in China several times. The topic “Yunnan University Cafeteria Launches Rose Banquet” has been on the Weibo hot search list many times, with 200 million readers and thousands of people actively participating in the discussion, generating positive social and emotional identification effects. By integrating Yunnan’s diverse culture and customs, historical heritage, and cultural accumulation with a European architecture and landscape environment, Yunnan University has formed an “elegant, refined, and scenic” edible campus, which is well-known throughout the country [50]. Edible landscapes at this scale and level of completeness at other universities are relatively scarce. The Damascus Rose Garden, Lavender Garden, and Vineyard provide laboratory space for students of different disciplines.

Yunnan University has built sewage treatment projects and fertilizer treatment stations, and it approves water usage according to different vegetation types such as trees, shrubs, and grasslands. The university has achieved recycling and zero discharge of campus wastewater, making efficient use of existing resources. It has formed characteristic landscape planning based on three axes (ritual axis, celebration axis, scholarly axis), three rings (outer ring, middle ring, inner ring), three gardens (scholar garden, star garden, moon garden), and three districts (teaching park, life park, sports park).

In addition to Yunnan University, other universities in the university town also have edible plants of different scales. Chenggong University Town is rich in edible landscape vegetation types, which can be divided into edible, medicinal, and both edible and medicinal types (Table 1). The rose garden covers an area of 21.46 acres and is in full bloom in April and May every year, producing approximately 4 to 5 tons of roses, which are used to make rose dishes, flower cakes, and rose oil. There are approximately 2579 Baozhu pear trees, which mature in early October each year, with an annual output of 600 to 800 kg. The roses and Baozhu pear provide the university with a special agricultural industry system, bringing direct economic benefits. Therefore, empirical research using Chenggong University Town as a case study has a certain representativeness.

### 3.2. Data Type, Source, and Method of Data Collection

The data were obtained from field research and questionnaires. The questionnaire was determined by modifying a commonly used standard questionnaire to make it more in line with research goals. The questionnaire was based on the classic questionnaire about “perception-identification-behavioral intention” commonly used by most Chinese scholars; we selected three latent variables of perception, emotional identification, and behavioral intention in this article. In the latent variable of perception, we added observed variables of functional perception and beauty perception. In the latent variable of emotional identification, we added an observed variable of likeness. In the latent variable of behavioral intention, we added an observed variable of consumer behavior. Then, we designed specific item contents around the research objectives. We finalized the questionnaire based on the above.

The respondents were teachers and students in Chenggong University Town, and a random sampling principle was used. Respondents were selected through random visits and sampling in various campuses in Chenggong University Town, and they filled out the questionnaire through one-to-one inquiries.

To obtain high-quality and effective questionnaire data, a pre-survey was first conducted. Sixty pre-study questionnaires were randomly distributed and collected, and the pre-study data were tested using SPSS 22.0 (IBM, Armonk, NY, USA). Cronbach’s α value was 0.961 and the Kaiser–Meyer–Olkin (KMO) value was 0.860, which was significant at the level of 0.000. Thus, the scale design was scientific and effective. Combined with the factor analysis of the pre-survey questionnaire, some items were further optimized to form a formal questionnaire.

The formal questionnaire consists of three parts. The first part is demographic characteristics. The second is the campus edible landscape experience, and the third is the emotional identification scale (Table 2). Because of the different blooming seasons for the different types of campus edible landscapes in Chenggong University Town, the formal questionnaire survey was conducted in Chenggong University Town on five days from July to December 2021 (20 July, 25 August, 27 September, 20 October, and 5 December). A total of 460 questionnaires were distributed, and all were collected, excluding six invalid questionnaires with incomplete answers. A total of 454 valid questionnaires were obtained, with an effective rate of 98.7%. 

Cronbach’s α in the overall scale is 0.935 and the KMO is 0.932, which is significant at the level of 0.000 (Table 3), indicating that the overall scale has a good level of reliability and validity. The sample data is suitable for factor analysis.

This study used principal component analysis to perform exploratory factor analysis and we extracted three components with feature values greater than 1, namely, emotional identification, perception, and behavioral intention. The variance loadings are 23.67%, 18.48%, and 17.48%, respectively, and the contribution rate of the cumulative explained variance is 59.62%. Among them, there is a factor load in the maximum variance rotation analysis that is greater than 0.5 in both perception and behavioral intentions. Excluding this item, the results show that all meet the load requirements.

### 3.3. An Improved SEM Method

The structural equation modeling (SEM) method is a multivariate statistical analysis method used to verify the influence relationship between independent variables and dependent variables [51,52,53]. This method involves the steps of structural model setting, parameter estimation, and evaluation of model fit degree [54], which is applicable to human geography, psychology, sociology, and other disciplines [55].

Therefore, this research aimed to improve the SEM method in terms of case-site selection and model validation. In the case study, a typical university town in China with a high reputation should be chosen as a typical case. Thus, in this research, Chenggong University Town in Kunming, Yunnan Province, was chosen as a case study. In terms of model validation, first, index factors were selected to construct the initial theoretical model of emotional identification from the three dimensions of perception, emotional identification, and behavioral intention, and hypotheses were formulated on the relationships among the relevant variables. Second, the SEM method was used to explore the fit validity of each factor in each dimension, perform standardized path analysis, verify the hypothesis results and the rationality of the initial model, and carry out model correction. Finally, a theoretical model of affective identification was constructed to explain the hypothetical paths and analyze the internal mechanisms and causes (Figure 1).

### 3.4. Initial Model

When teachers and students visit the edible landscape area of campus, they form a preliminary perception of the layout, beauty, and richness of the edible landscape. The campus environment, infrastructure conditions, public security, and services further affect the perceptions of teachers and students and determine the function and value of campus edible landscapes. The more positive the teachers and students’ perceptions of campus edible landscapes, the more likely they are to produce positive emotional identification. Based on this, we hypothesized H1.

**Hypothesis 1** **(H1).**
*Perception significantly and positively affects emotional identification.*


The environmental protection identification, service identification, liking degree, attachment feeling, and satisfaction degree of the edible landscape are the main contents of emotional identification. Positive emotional identification promotes positive behavioral intentions [56,57]. Based on this, we hypothesized H2.

**Hypothesis 2** **(H2).**
*Emotional identification significantly and positively affects behavioral intention.*


In addition, the perceptions generated by teachers and students in the process of visiting and experiencing the edible landscape of the campus may directly affect the behavioral intention to protect the environment, the behavioral intention to consume campus-produced food, and the behavioral intention to give friendly recommendations about the edible landscape [58]. Based on this, we hypothesized H3.

**Hypothesis 3** **(H3).**
*Perception significantly and positively affects behavioral intention.*


Based on the internal attributes, external scale, and quantity of edible landscape resources on campus, the perception dimension was constructed through nine indicators: function, culture, brand, richness, beauty, service, quality, environment, and layout. Combined with the characteristics of teachers and students’ emotional identification, the emotional identification dimension was constructed through five indicators: environmental protection identification, service identification, likeness, attachment, and satisfaction. From the perspective of teachers and students’ experiences, the behavior intention dimension was constructed through three indicators: protective behavior, consuming behavior, and friendly referral behavior. Based on the value characteristics of edible landscape resources on campus, this study established an initial model of emotional identification (Figure 2).

## 4. Results

### 4.1. Demographic Characteristics of the Sample

In the sample of teachers and students surveyed, women (61.3%) accounted for slightly higher proportions than men. Most of them were unmarried (92.8%). The sample covered 16 ethnic groups, of which the “Han” sample accounted for 87.4%. The age structure of the sample was dominated by 18 to 25 year-olds, accounting for 88.5% of the total sample, and the occupation was dominated by “full-time students from all provinces in China,” accounting for 91.2%. The overall educational level of the sample was relatively high, with “undergraduate” accounting for 73.5%, “master’s degree and above” accounting for 23.4%, and “undergraduate and below” accounting for 3.1%. The sample as a whole conformed to the principle of random sampling, and the reliability of the questionnaire was high.

### 4.2. Model Modification and Verification

In this study, a confirmatory factor analysis was used to test the rationality of each impact factor and the fit of the theoretical model, and the variables of each dimension were adjusted according to the results of the confirmatory factor analysis. Because four observation variables—layout perception (GZ5), service perception (GZ7), quality perception (GZ8), and environmental identification (RT1)—had large correction modification index (MI) values, and the standardized path coefficient was not ideal, the minimum discrepancy per degree of freedom (CMIN/DF), adjusted goodness-of-fit index (AGFI), normed fit index (NFI), goodness-of-fit index (GFI), incremental fit index (IFI), comparative fit index (CFI), and root mean square error of approximation (RMSEA) values of the initial assumption model did not reach the adaptation range (Table 4). Furthermore, we attempted to increase the path of mutual influence between variables with larger MI values and explore the relationship between some observed variables. The adjusted standardized path coefficient was not significant, indicating that the correlation between GZ5, GZ7, GZ8, and RT1 was not obvious, and the influence of the four observed variables was not significant. Thus, we deleted GZ5, GZ7, GZ8, and RT1. Our field research revealed that, owing to the good environmental protection of Chenggong University Town, teachers and students pay less attention to the layout, service, and quality of the edible landscape on campus, and the perception is weak. The environmental protection recognition is not strong, and the impact of GZ5, GZ7, GZ8, and RT1 is very small.

After adjusting the observation variables, the CMIN/DF value of the modified model was reduced to the adaptation range, the AGFI, NFI, GFI, IFI, and CFI values were greater than 0.9, and the RMSEA value also reached the adaptation range. The fit degree of the emotional identification model after the above correction reached a fitting state.

In this study, confirmatory factor analysis was performed on the latent variables of the revised model (Table 5). The critical ration (CR) values of the combined reliability of the three dimensions were all greater than 0.7, and the combined reliability was good. The average variance extracted (AVE) value of the convergent validity of the emotional identification and behavioral intention dimensions were both greater than 0.5, and the convergent validity was excellent. The convergent validity of the perception was between 0.3 and 0.5, which is acceptable. In summary, the model has convergent validity. The revised model and standardized path coefficients are shown in Figure 3.

First, the *p*-value of perception on emotional identification and emotional identification on behavioral intention reached a very significant level. H1 and H2 were verified because the results show that perception significantly and positively affects emotional identification, and emotional identification significantly and positively affects behavioral intention. Second, the *p*-value of emotional identification on behavioral intention reached a significant level. H3 was verified, and perception significantly and positively affects behavioral intention (Table 6).

## 5. Discussion

### 5.1. Surface Value Affects Perception

By combining the revised emotional identification model with the value of campus edible landscape resources and analyzing its internal connection, the study demonstrates that campus edible landscapes have a dual value. The first level is the surface value, that is, popular viewing and recreation, scientific research and education, ecology and sustainability, and economic benefits, which are conventional values. The second level is the deep value, that is, emotional identification, which is a derivative value.

The value of recreation significantly affects teachers and students’ perceptions of the richness, beauty, and environmental perception of campus edible landscapes. The value of scientific research and education shapes functional, cultural, and brand perceptions. The economic value attached to the campus edible landscape is conducive to the formation of campus brand perception. Ecology and sustainability values further enhance the perception of richness, beauty, environment, and function. Therefore, in the process of teacher-student interaction, the superficial value of campus edible landscapes affects the perception (Figure 4).

### 5.2. Perception Influences Deep Value (Emotional Identification)

Teacher–student emotions have individual and complex characteristics of changeable time processes, and their understanding of campus edible landscapes has its advantages, leading to the complexity and instability of emotional identification. Teachers and students’ positive emotional identification is composed of service identification, favor, attachment, and satisfaction. Because perception extremely and significantly affects emotional identification, the standardization coefficients of functional perception, richness perception, and beauty perception are higher than 0.7 in the verification of the emotional identification SEM, confirming that it is the main influencing factor of emotional identification. Therefore, although perception has a direct and significant driving effect on behavioral intention, the path of “perception influences emotional identification, and emotional identification influences behavioral intention” is more significant. More precisely, emotional identification plays a highly significant mediating role between perception and behavioral intention.

The surface value serves as the basic condition for the formation of deep value (emotional identification). The surface value affects teachers and students’ perception in the “emotion-landscape” interaction, thereby affecting the emotional identification of teachers and students. Emotional identification, as the connection point between perception and behavioral intention, forms a positive behavioral intention and promotes a virtuous circle in which surface values influence perception, perception influences deep value (emotional identification), and deep value (emotional identification) influences behavioral intention.

Based on the association mechanism model of campus edible landscape resources and emotional identification, a virtuous cycle model of “emotional identification-protection” is formed. The association mechanism model fully reflects the special emotional effect between the surface value and the deep value of campus edible landscapes. It also highlights the function of campus edible landscapes serving teachers and students and the effect of “emotion-landscape” integration and symbiosis. This research enhances campus identification and provides a scientific basis for subsequent studies on the value of campus landscape resources.

## 6. Conclusions

During model modification and verification, the influence of some variables in the perception and identification dimensions is unstable, such as layout perception, service perception, quality perception, and environmental protection identification. Some variables have volatility and need to be deleted based on the actual situation to obtain the best model. The association mechanism and influence path in the emotional identification model based on the improved SEM method are typical, representative, and applicable to Chinese universities.

Hypothetical paths H1 and H2 are more significant than H3. Thus, teacher–student emotional identification plays an intermediary role between the perception and behavioral intention of campus edible landscapes, manifested as the extremely significant “perception influences emotional identification, and emotional identification influences behavioral intention” effect and a significant “perception influences behavior intention” driving path.

In the “emotion-landscape” interaction, the key scientific issue of emotional identification research from the perspective of campus edible landscapes is the association mechanism in which surface values influence perception, perception influences emotional identification, and emotional identification influences behavioral intention. Surface values, such as recreation, ecology and sustainability, economic benefit, and research and education, are the basic conditions for generating emotional identification.

Looking forward to the research progress of campus edible landscapes and emotional geography, domestic and foreign research on edible landscapes has focused primarily on planning and design. The attempt to explore the emotional identification mechanisms of campus edible landscapes from the perspective of emotional geography is relatively new. From the perspective of university towns, this study enriched the research content of edible landscapes by exploring the emotional identification mechanisms of campus edible landscapes. At the same time, this study was also an exploration of the research paradigm regarding emotional geography with the hope of increasing the importance of human emotion and landscape research.

Regarding practical application, universities should adapt measures to local conditions and improve plans to create a participatory campus edible landscape base. Relying on the natural and cultural resources and infrastructure of the campus, the habitats of edible plants should be fully considered, and abandoned land should be fully used to plant ornamental edible plants to serve teachers and students. This will enhance the emotional identification of teachers and students and improve the efficiency of land use. Combined with the function and value of campus edible landscapes, students can participate in the process of planning, planting, cultivating, and picking edible plants, and they can help to make food from them. Students can also put forward their own suggestions and ideas for the construction of campus edible landscapes. In the whole process, students gain a sense of happiness, which enhances their sense of belonging and attachment to the university. Thus, it is helpful to cultivate students’ practical abilities and, through training, this can create a good emotional effect and build the characteristic college brand.

## Figures and Tables

**Figure 1 ijerph-19-11425-f001:**
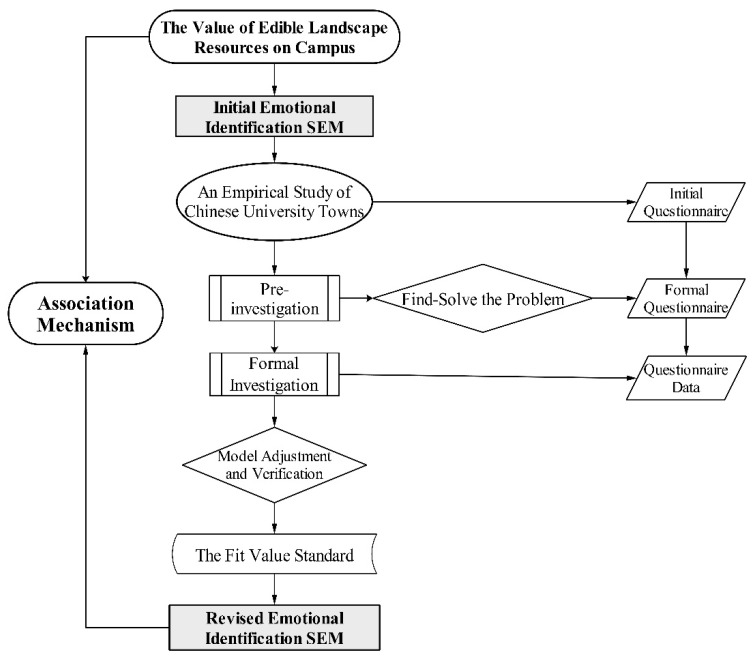
SEM flow chart.

**Figure 2 ijerph-19-11425-f002:**
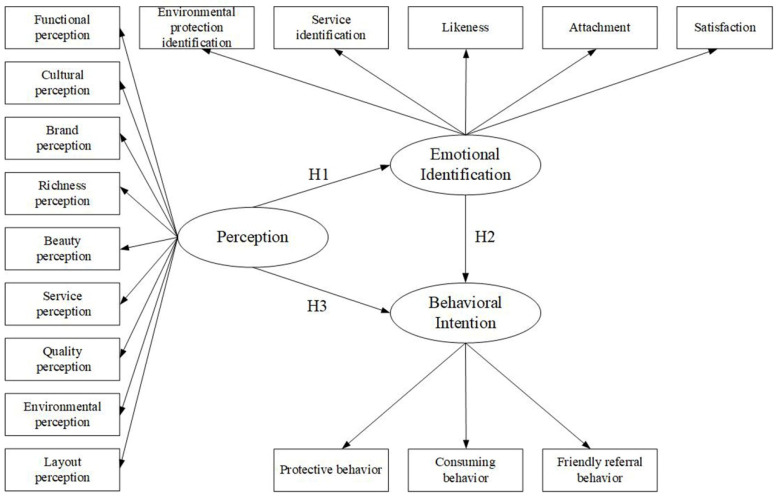
Hypothetical theoretical model.

**Figure 3 ijerph-19-11425-f003:**
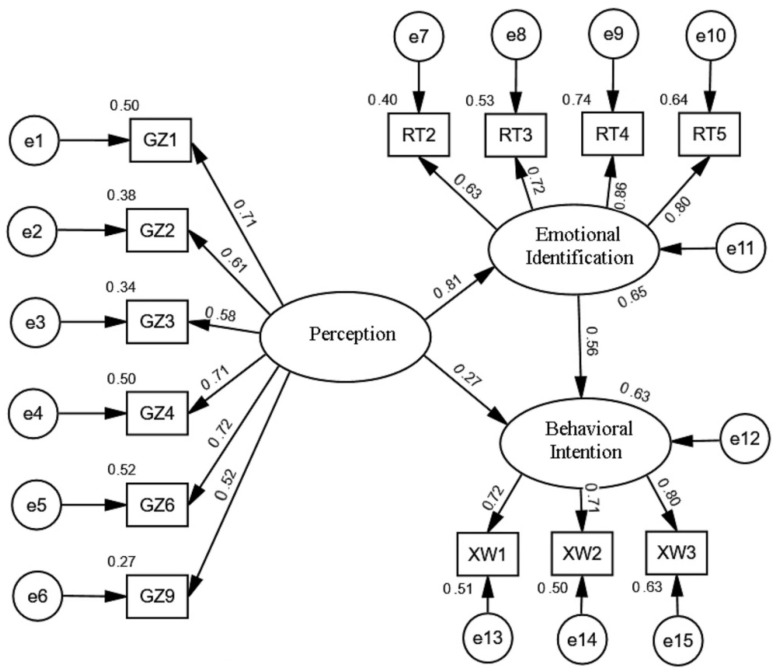
Revised emotional identification model of the edible landscape on campus.

**Figure 4 ijerph-19-11425-f004:**
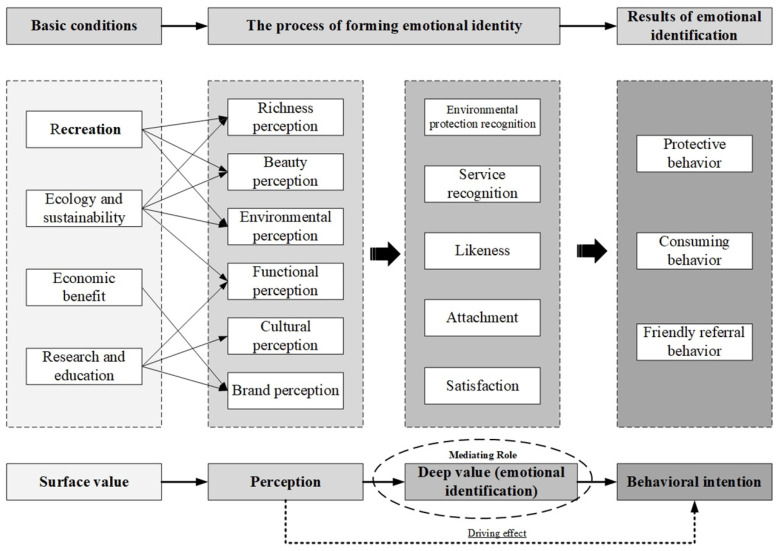
The association mechanism model.

**Table 1 ijerph-19-11425-t001:** Types of edible landscape plants in Chenggong University Town.

Edible Types	Plant Species
Edible	Baozhu pears, Grapes, Oranges, Osmanthus fragrans ‘Latifolius’, Magnolia liliflora Desr., Cedar, Docynia delavayi, Musa basjoo Siebold, Ivy, Ophiopogon, Pumpkin, Kiwi, Houttuynia cordata, Persimmon, Bayberry.
Medicinal	Kunming Pittosporum tobira, Cornus kousa subsp, Sapindus, Cinnamomum camphora (L.) Presl., Deyeuxia langsdorffii (Link) Kunth, Itoa orientalis Hemsl, Erythrina variegata Linn, Magnolia delavayi Franch, Jacaranda mimosifolia D. Don, Ficus concinna Miq, Cinnamomum japonicum Sieb, Manglietia insignis, Acer truncatum Bunge, Metasequoia, Michelia alba DC., Ilex microphyllam, Photinia serratifolia (Desfontaines) Kalkman, Camptotheca acuminata, Alnus cremastogyne, Casuarina equisetifolia L., Callistemon rigidus R. Br., Hibiscus rosa-sinensis, Cycas revoluta, Hawthorn, Ficus tikouaBur., Gypsophila paniculata L., Arrhenatherum elatius f. variegatum, Chlorophytum comosum (Thunb.) Baker., Viola phillipina, Ligustrum quihoui Carr, Rosmarinus officinalis, Sapium sebiferum (L.) Roxb.
Both edible and medicinal	Rosa rugosa Thunb, Lavender, Prunus paracerasus, Ginkgo, Rosa chinensis Jacq, Osmanthus fragrans, Leonurus japonicus Houtt, Mentha haplocalyx, Pistacia chinensis Bunge, Chrysanthemum, Chimonanthus praecox, Canna indica L., Prunus cerasifera f. atropurpurea, Lagerstroemia indica L., White magnolia, Catalpa ovata G. Don, Loquat, Schisandra chinensis, Cercis chinensis, Lonicera japonica Thunb.

**Table 2 ijerph-19-11425-t002:** Questionnaire scale.

Latent Variables	Observed Variables and Codes	Item Contents
Perception	GZ1 Functional perception	Campus edible landscape has the functions of leisure, stress relief, and mood adjustment. Campus edible landscape has the functions of knowledge learning, science popularization, and skill training. Campus edible landscape has the function of bringing economic benefits to the university. Campus edible landscape has the function of improving the ecological environment. Campus edible landscape has the function of promoting scientific research and teaching.
	GZ2 Cultural perception	Campus edible landscape is vital to the construction of campus culture.
	GZ3 Brand perception	Campus edible landscape has unique brand value.
	GZ4 Richness perception	Chenggong University Town has rich and unique edible landscapes.
	GZ5 Layout perception	The layout and planning of the campus edible landscape are reasonable.
	GZ6 Beauty perception	Campus edible landscape is beautiful.
	GZ7 Service perception	Variety of campus services, security thoughtful, and good service attitude. Price of edible landscape series foods is reasonable. Edible landscape series foods are rich in variety and delicious.
	GZ8 Quality perception	The campus has good supporting infrastructure and environmental sanitation.
	GZ9 Environmental perception	The environment in and around the campus is beautiful and the air is fresh.
Emotional identification	RT1 Environmental protection identification	I agree with protecting campus edible landscapes.
	RT2 Service identification	I agree with the service philosophy and public services.
	RT3 Likeness	I love edible landscapes more than other types of landscapes. I love the edible landscape of Chenggong University Town more than other places.
	RT4 Attachment	When I look at edible landscapes, I am in a pretty good state of mind and feel very happy. When I look at edible landscapes, my spiritual needs are easily met. I have become obsessed with the edible landscape.
	RT5 Satisfaction	I am very satisfied with visiting and experiencing the edible landscape on campus. I am very satisfied with the edible landscape series of food.
Behavioral intention	XW1 Protective behavior	I would like to participate in the preservation of the campus edible landscape environment and culture
	XW2 Consumer behavior	I would like to buy food and daily necessities related to the edible landscape on campus.
	XW3 Friendly referral behavior	I would like to promote and praise the edible landscape in Chenggong University Town and recommend others to come.

**Table 3 ijerph-19-11425-t003:** Reliability and validity test of scale.

Variable	N	Cronbach’ s α	KMO	Sig.
Perception	15	0.880	0.885	0.000
Emotional identification	9	0.895	0.883	0.000
Behavioral intention	3	0.785	0.706	0.000
Overall scale	27	0.935	0.932	0.000

**Table 4 ijerph-19-11425-t004:** Confirmatory factor analysis model fitting comparison.

Fit Index	CMIN/DF	RMR	AGFI	NFI	GFI	IFI	CFI	RMSEA
Adaptation value	1–3	<0.05	>0.9	>0.9	>0.9	>0.9	>0.9	<0.08
Initial model	4.192	0.035	0.827	0.863	0.870	0.892	0.870	0.087
Revised model	2.806	0.030	0.911	0.927	0.939	0.952	0.952	0.066

**Table 5 ijerph-19-11425-t005:** The revised confirmatory factor analysis.

Latent Variable	Observed Variable	Parameter Significance Estimation				Factor Loading	Topic Reliability	Combination Reliability	Convergence Validity
		Unstd.	S.E.	T-Value	*p*	Std.	SMC	CR	AVE
Perception	GZ1 Functional perception	1.000				0.735	0.540	0.809	0.418
	GZ2 Cultural perception	1.232	0.107	11.502	***	0.636	0.404		
	GZ3 Brand perception	1.203	0.110	10.946	***	0.601	0.361		
	GZ4 Richness perception	1.548	0.126	12.278	***	0.698	0.487		
	GZ6 Beauty perception	1.411	0.114	12.388	***	0.689	0.475		
	GZ9 Service perception	0.883	0.098	8.962	***	0.490	0.240		
Emotional identification	RT2 Service identification	1.000				0.601	0.361	0.842	0.575
	RT3 Favor	1.263	0.106	11.882	***	0.746	0.557		
	RT4 Attachment	1.217	0.099	12.244	***	0.894	0.799		
	RT5 Satisfaction	1.123	0.095	11.866	***	0.764	0.584		
Behavioral intention	XW1 Protective behavior	1.053	0.089	11.898	***	0.733	0.774	0.874	0.699
	XW2 Consumer behavior	1.062	0.089	11.895	***	0.732	0.774		
	XW3 Friendly recommendation behavior	1.000				0.763	0.548		

*** *p* < 0.001 (very significant).

**Table 6 ijerph-19-11425-t006:** Model path and verification.

Hypothetical Path	Standardization Factor	*p* Value	Verification Result
H1: Perception influences emotional identification	0.81	***	very significant
H2: Emotional identification influences behavior intention	0.56	***	very significant
H3: Perception influences behavior intention	0.27	0.008 **	significant

** *p* < 0.01 (significant), *** *p* < 0.001 (very significant).

## Data Availability

The data presented in this study are available on request from the corresponding author.

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
