# Peer review of "Analysis of the Emotional Identification Mechanism of Campus Edible Landscape from the Perspective of Emotional Geography: An Empirical Study of a Chinese University Town"

_ijerph, 2022, doi:10.3390/ijerph191811425_

Round 1
Reviewer 1 Report
Thank you for allowing me the opportunity to read your interesting paper. I do hope these comments will help you in the review process.
In framing the paper some further interplay with relevant literature would be helpful. The notion and practice of edible landscape is more than a multifunctional landscaping method. It refers to food focused urbanism and has been subject of research for a considerable number of years. The authors should explore the food and urbanism literature more broadly and see sources such as the Routledge Handbook of Landscape and Food edited by Zeunert and Waterman (2018) and articles in this area in the Journal of Urbanism for example. This wider exploration of relevant literature will help to situate the rather narrow, instrumental approach suggested here more adequately. Pastoral cities were conceptualised far earlier than the 1970s. See for example garden city traditions and practices and the post Second World War university campus-new town model. There is a substantial literature on urban agriculture so notions like ‘rational development of edible landscapes’ need to be explained i.e., what constitutes ‘rational’? why is this important? and to be situated in relation to this too. Again, the authors could delve into these antecedents and themes to ground their paper in relevant literatures.
If it is intended to make claims such as “In the 1970s, the modern pastoral city concept became the trend in landscape design and urban planning.’ and “Relying on a well-organized urban agricultural system, it has formed developed representative edible landscape models such as rooftop farms and vertical farms” (as examples) the authors do need to cite evidence in support. This needs to be done throughout to support claims.
It is noted that ‘In the 1990s, the ecological campus project was initiated in China.” but not made clear what ‘the’ ecological campus project was or is - is there only one? where is it? what kinds of campus landscapes (design, placemaking aspects, geographical, governance locations) does it occur in Source citations are again needed. What constitutes an edible campus landscape needs to be more clearly defined. Does this shift with the spatial nature of campuses - are urban, suburban, rural and conurbation campus different? Are high density locations different in edible terms from lower density ones?
Some key constructs need unpacking - what is emotional identification? What is formal identification? There should be some discussion of key theorists who shaped notions of place attachment and identity from the work of Tuan onwards to situate this.
Before moving into the methodology, and the details of any modelling based on data collected, the authors also need to provide a reflexive rationale for why edible landscapes should be treated as an instrumental method for meeting emotional needs or contributing to tourism branding (if the arguments are accepted that they can do so). The additional justifications that seem to emerge in the final paragraph of students’ practical ability and thinking training and campus branding need to be worked in here too. This will provide a more developed rationale and justification for the research.
It is intriguing that in a study focused on food taste and conviviality related to the edible produce itself are not explored. It’s noted that these edible landscapes are ornamental and for interacting with scenery. Why is that the case? Why are these not actual food sources? (if that is a correct reading). Does this mean they are not in fact edible? There could be a connection to gastronomic landscape research and analysis which might help balance the instrumental approach that seems dominant here in which food is apparently viewed as a tool for other ends rather than a way to develop a convivial ecology and localise food systems (another area that could so with some coverage here)?
Could the discussion of beauty be connected to a more developed spatial design analysis of landscape? Could some sense of these locations as landscaped places be developed through description of the sites and their variation to see what impact these urbanism points have on perception and identification? This might be an area to review.
There are some issues with referencing of putting first names instead of last names in the citation. For example, Amanda should be Kearney (last name) A (initial of Amanda which is first name). Similarly with Parr and other scholars, the first and last names are confused in the references. This needs review and correction throughout the paper.
Citation needs to meet academic requirements for quoted material. Examples like this need to be properly cited: “By integrating Yunnan’s diverse culture and customs, historical heritage and cultural accumulation, and European architecture and landscape environment, Yunnan University has formed an “elegant, refined, and scenic” edible campus, which is well-known throughout the country.” As there is direct quotation author, date and page reference are all needed.
In summary this is a very interesting topic well supported by a large scale survey but one that might benefit from considering the points made above to situate the work, provide a convincing rationale and justification for aims that are clarified, and explore some of the intriguing aspects that are currently gaps in the discussion and analysis.
Author Response
Response to Reviewer 1 Comments
Dear reviewer,
We appreciate the time and effort you have put in to reviewing our manuscript (number: ijerph-1846139). We thank you for your detailed comments, which have helped us improve the quality of this paper. In the following text, we address your concerns in the order they were provided.
Sincerely,
Qi Yi
Point 1: In framing the paper some further interplay with relevant literature would be helpful. The notion and practice of edible landscape is more than a multifunctional landscaping method. It refers to food focused urbanism and has been subject of research for a considerable number of years. The authors should explore the food and urbanism literature more broadly and see sources such as the Routledge Handbook of Landscape and Food edited by Zeunert and Waterman (2018) and articles in this area in the Journal of Urbanism for example. This wider exploration of relevant literature will help to situate the rather narrow, instrumental approach suggested here more adequately. Pastoral cities were conceptualised far earlier than the 1970s. See for example garden city traditions and practices and the post Second World War university campus-new town model. There is a substantial literature on urban agriculture so notions like ‘rational development of edible landscapes’ need to be explained i.e., what constitutes ‘rational’? why is this important? and to be situated in relation to this too. Again, the authors could delve into these antecedents and themes to ground their paper in relevant literatures.
Response 1: After careful consideration, we have addressed your insightful comments by making the following modifications to the article.
- In the introduction, we have added a description of the importance of landscape and food, and we have cited the Routledge Handbook of Landscape and Food. We have also added other relevant references.
- As you note, the concept of a garden city originated earlier, but the documents we found showed that it was an urban design concept in the 19th century, so “1970s” was changed to “19th century.”
- In the introduction, we have added a discussion of the rational development of the edible landscape.
Point 2: If it is intended to make claims such as “In the 1970s, the modern pastoral city concept became the trend in landscape design and urban planning.’ and “Relying on a well-organized urban agricultural system, it has formed developed representative edible landscape models such as rooftop farms and vertical farms” (as examples) the authors do need to cite evidence in support. This needs to be done throughout to support claims.
Response 2: Based on your suggestion, we have added citations for the corresponding references in both parts of the article.
Point 3: It is noted that ‘In the 1990s, the ecological campus project was initiated in China.” but not made clear what ‘the’ ecological campus project was or is - is there only one? where is it? what kinds of campus landscapes (design, placemaking aspects, geographical, governance locations) does it occur in Source citations are again needed. What constitutes an edible campus landscape needs to be more clearly defined. Does this shift with the spatial nature of campuses - are urban, suburban, rural and conurbation campus different? Are high density locations different in edible terms from lower density ones?
Response 3: The ecological campus project was a campus construction concept proposed in China, and specific projects and plans were implemented by each university. We proposed China's ecological campus project in the introduction, which is aimed at eliciting an edible landscape development basis in Chinese university towns.
Based on your suggestion, we added a definition to the campus edible landscape in the first paragraph of 2.1, and we have differentiated campus edible landscapes from edible landscapes in cities, suburbs, and other places in the second paragraph of section 2.1. They differ mainly in size and function. Campus edible landscapes often need to have a certain scale, and some low-density and scattered edible plants are not in the category of ornamental and cultivation. Relevant supplements have been added to section 2.1.
Point 4: Some key constructs need unpacking - what is emotional identification? What is formal identification? There should be some discussion of key theorists who shaped notions of place attachment and identity from the work of Tuan onwards to situate this.
Response 4: Based on your suggestion, we started from Tuan 's concept of place. We then mentioned emotional geography and emotional identification to pave the way for the concept of emotional identification. The relevant content is supplemented in section 2.2.
Point 5: Before moving into the methodology, and the details of any modelling based on data collected, the authors also need to provide a reflexive rationale for why edible landscapes should be treated as an instrumental method for meeting emotional needs or contributing to tourism branding (if the arguments are accepted that they can do so). The additional justifications that seem to emerge in the final paragraph of students’ practical ability and thinking training and campus branding need to be worked in here too. This will provide a more developed rationale and justification for the research.
Response 5: Based on your suggestion, the following modifications have been made to the article.
- In the introduction, we added the reasons that edible landscapes can meet the emotional needs of teachers and students or promote tourism branding.
“Since the teaching activities are held on campus from Monday to Friday, teachers and students have more opportunities to contact the campus landscape, especially edible landscapes. Campus edible landscapes integrate viewing, eating and enjoyment, and they provide teachers and students with an opportunity for human–land interaction. Participating in the interaction of campus edible landscapes can meet students and teachers’ emotional needs for sightseeing, work, and rest, and it can help with mood adjustment. In this context, universities in various university towns have pro-moted the construction of campus edible landscapes to create more well-known campus tourism brands.”
- In the conclusion, we also added supplementary reasons in the last paragraph.
“Combined with the function and value of campus edible landscapes, students can participate in the process of planning, planting, cultivating, and picking of edible plants, and they can help to make food from them. Students can also put forward their own suggestions and ideas for the construction of campus edible landscapes. In the whole process, students gain a sense of happiness, which enhances their sense of belonging and attachment to the university. Thus, it is helpful to cultivate students’ practical ability and though training; this can create a good emotional effect, and build the characteristic college brand.”
Point 6: It is intriguing that in a study focused on food taste and conviviality related to the edible produce itself are not explored. It’s noted that these edible landscapes are ornamental and for interacting with scenery. Why is that the case? Why are these not actual food sources? (if that is a correct reading). Does this mean they are not in fact edible? There could be a connection to gastronomic landscape research and analysis which might help balance the instrumental approach that seems dominant here in which food is apparently viewed as a tool for other ends rather than a way to develop a convivial ecology and localise food systems (another area that could so with some coverage here)?
Response 6: In the article, we did not address campus edible landscapes as part of the local food system or a way to develop convivial economy for the following reasons:
- Campus edible landscapes in Chinese University Town is mainly for the teachers and students, not for profit. Because the planting costs and losses are subsidized by universities, fruits produced are provided for free picking and tasting by teachers and students, and the processed food is also sold in the cafeteria at an ultra-low price.
- Campus edible landscapes are independent of the local food system. The management of universities is not bound by the relevant local agricultural departments, and has a certain degree of independence and autonomy.
In this study, content related to food taste and edible produce is mainly included in the questionnaire items, and they are investigated as aspects of the perception dimension.
Point 7: Could the discussion of beauty be connected to a more developed spatial design analysis of landscape? Could some sense of these locations as landscaped places be developed through description of the sites and their variation to see what impact these urbanism points have on perception and identification? This might be an area to review.
Response 7: Thank you very much for your comment. Your suggestion is something we hadn't considered before. In follow-up research, we believe that we can try to strengthen the discussion of its connection with landscape space design, landscape place perception, and other aspects.
Point 8: There are some issues with referencing of putting first names instead of last names in the citation. For example, Amanda should be Kearney (last name) A (initial of Amanda which is first name). Similarly with Parr and other scholars, the first and last names are confused in the references. This needs review and correction throughout the paper.
Response 8: We have corrected the references and ensure that the authors’ names are cited in the correct order.
Point 9: Citation needs to meet academic requirements for quoted material. Examples like this need to be properly cited: “By integrating Yunnan’s diverse culture and customs, historical heritage and cultural accumulation, and European architecture and landscape environment, Yunnan University has formed an “elegant, refined, and scenic” edible campus, which is well-known throughout the country.” As there is direct quotation author, date and page reference are all needed.
Response 9: We have corrected the citations throughout the article.
Point 10: In summary this is a very interesting topic well supported by a large scale survey but one that might benefit from considering the points made above to situate the work, provide a convincing rationale and justification for aims that are clarified, and explore some of the intriguing aspects that are currently gaps in the discussion and analysis.
Response 10: Thank you very much for your comment. Your review comments have been of great help in improving and revising this article. Based on your suggestions, we have included additional arguments to the article. We hope our revision makes the article clearer so that more people can understand it.

Reviewer 2 Report
The title is promising, yet the content of the paper needs reconsidering and re-writing.
The authors did the extended mathematical work to prove the structural equation modeling validity in the field of the emotional geography. But in this study the elaborated statistics confirm the obvious, resulting from the logic, facts.
It is the case study of one of Chinese universities, and not mediocre, but the one that has been several times awarded for its beautiful campus. This fact, however, does not make it a representative of all Chinese universities. The study lacks therefore national and international perspective.
Also the weak point of the study is the survey itself. The authors write about ‘field research’ performed (line 180) but they do not address them later in the text. Also the methodology of questionnaire is not clear (see comment below).
As far as 91,2% of the surveyed population declared themselves as ‘full-time students’ the study results present mainly student’s but not teacher’s point of view. It would be interesting to analyze these two groups separately as far as e.g. a student spends 4-6 years in the university campus while teacher usually more.
There is also no mention on whether the Chenggong University is accessible also for foreign students and if so, they were included or excluded from the study, because related cultural differences may be confounding factor e. g. in cultural or beauty perception.
The paper also needs structure correction– the aim of the study and conclusion are unclear.
I attach below some suggestions which may help to improve the paper.
57-66 This section belongs rather to ‘Aim’.
116-123 Is this section relevant to the study? Have the Holocaust case studies and children social benefit something in common with the edible landscape of Chinese universities?
176-177 I would rather say that Chenggong University is still a model university as far as an edible landscape is concerned but this idea is not yet so developed in other Chinese universities (as mentioned in lines 158-159). Therefore it can not be called ‘highly representative’.
241-277 This section belongs rather to ‘Introduction/state of research’.
189-195 How was the questionnaire constructed, who is its author – one of the researchers? a sociologist? Was it custom made or was it one of commonly available standard questionnaires modified for the study purpose?
When the survey took place – on five days chosen from 6 months? Does it means that some respondents visited the landscape in summer and some in winter?
How were the respondents recruited – were they volunteers who have answered some advert or have they been asked personally?
215-226 This section belongs rather to ‘Introduction/state of research’.
227-238 This section belongs rather to ‘Aim’.
242-277 This section belongs rather to ‘Introduction/state of research’.
Figure 4. The part describing which ‘basic condition’ from the ‘surface value’ influences which perception type may be interesting, but unfortunately is not highlighted in the results and conclusions.
Author Response
Response to Reviewer 2 Comments
Dear reviewer,
We appreciate the time and effort you have put in to reviewing our manuscript (number: ijerph-1846139). We thank you for your detailed comments, which have helped us improve the quality of this paper. In the following text, we address your concerns in the order they were provided.
Sincerely,
Qi Yi
Point 1: The title is promising, yet the content of the paper needs reconsidering and re-writing.
The authors did the extended mathematical work to prove the structural equation modeling validity in the field of the emotional geography. But in this study the elaborated statistics confirm the obvious, resulting from the logic, facts.
Response 1: Your comments are very thoughtful and helpful. We have carefully reviewed the paper, and we believe that the revised manuscript addresses your concerns.
Point 2: It is the case study of one of Chinese universities, and not mediocre, but the one that has been several times awarded for its beautiful campus. This fact, however, does not make it a representative of all Chinese universities. The study lacks therefore national and international perspective.
Response 2: We believe that Chenggong University Town, as one of the more mature university towns in China, has a certain representativeness for the construction of an edible landscape on campus. Our reasons are as follows:
- The campus is located in Kunming, China, which has a spring-like climate all year round and boasts abundant plant life. Kunming was even chosen to host this year's COP15. As a case study of edible landscape planting, we believe Chenggong University Town in Kunming can serve as a model for similar landscape plans.
- Chenggong University Town was built relatively recently, drawing on the early construction experience of other university towns in China.
- Of the university towns in China, Chenggong University Town has one of the better edible landscape plans, and it embodies certain school-running models with Chinese characteristics. Thus, it can help domestic and foreign scholars better understand China's university towns. At the same time, it can provide a reference for other university towns to plan edible landscapes.
Based on your suggestion, we have changed the title of the article to “Analysis of the Emotional Identification Mechanism of Campus Edible Landscape from the Perspective of Emotional Geography: An Empirical Study of a Chinese University Town.”
Point 3: Also the weak point of the study is the survey itself. The authors write about ‘field research’ performed (line 180) but they do not address them later in the text. Also the methodology of questionnaire is not clear (see comment below).
Response 3: Based on your suggestion, we have improved the method of questionnaire, and we have added some details about the survey in section 3.2 of the article.
Point 4: As far as 91,2% of the surveyed population declared themselves as ‘full-time students’ the study results present mainly student’s but not teacher’s point of view. It would be interesting to analyze these two groups separately as far as e.g. a student spends 4-6 years in the university campus while teacher usually more.
Response 4: In Chenggong University Town, the ratio of teachers to students is about 1:30. Therefore, students make up the vast majority of the survey sample. This is related to the ratio of teachers to students in Chenggong University Town, which affects the representativeness of the sample to a certain extent. That is, the sample mainly represents the views of students. However, we agree that it would be quite interesting and informative to analyze these two groups separately in future studies.
Point 5: There is also no mention on whether the Chenggong University is accessible also for foreign students and if so, they were included or excluded from the study, because related cultural differences may be confounding factor e. g. in cultural or beauty perception.
The paper also needs structure correction– the aim of the study and conclusion are unclear.
I attach below some suggestions which may help to improve the paper.
Response 5: The research, which excluded foreign students, was aimed at studying Chinese university students and teachers. In addition, based on your suggestion, we have summarized the research purpose in the last paragraph of the introduction section. We have also adjusted the structure of part of the paper and strengthened the conclusion section.
Point 6: 57-66 This section belongs rather to ‘Aim’.
Response 6: Based on your suggestion, we have put the research aim at the end of the introduction, and we have moved lines 57-66 in the original to the last paragraph of the introduction.
Point 7: 116-123 Is this section relevant to the study? Have the Holocaust case studies and children social benefit something in common with the edible landscape of Chinese universities?
Response 7: The Holocaust case study is designed to analyze the impact of Holocaust classroom cases on the emotional interaction between teachers and students, and thus it is related to the research theme of this article. However, children's social benefit is less relevant to this research, so we deleted “Children’s discursive expressions are associated with emotional change, and the inclusion of their ideas is a way to achieve equitable distribution of children’s social benefits (Fegter & Mock, 2019)” from the manuscript.
Point 8: 176-177 I would rather say that Chenggong University is still a model university as far as an edible landscape is concerned but this idea is not yet so developed in other Chinese universities (as mentioned in lines 158-159). Therefore it can not be called ‘highly representative’.
Response 8: Thank you very much for your comment. On line 272 of the revised article, we changed “highly representative” to “have a certain representativeness.”
Point 9: 241-277 This section belongs rather to ‘Introduction/state of research’.
Response 9: Based on your suggestion, we moved lines 241-277 of the original article to “2.1. Campus edible landscape.”
Point 10: 189-195 How was the questionnaire constructed, who is its author – one of the researchers? a sociologist? Was it custom made or was it one of commonly available standard questionnaires modified for the study purpose?
Response 10: After careful consideration, the following modifications have been made to the article.
- In the “3.2 Data type, source, and method of data collection” section, we added a detailed description of the source and construction of the questionnaire to the first paragraph.
“The data were obtained from field research and questionnaires. The questionnaire was determined by modifying a commonly used standard questionnaire to make it more in line with the purpose of this study. The respondents were teachers and students in Chenggong University Town, and a random sampling principle was used. Respondents were selected through random visits and sampling in various campuses in Chenggong University Town, and they filled out the questionnaire through one-to-one inquiries.”
- The questionnaire was determined by modifying a commonly used standard questionnaire to make it more in line with the purpose of this research.
Point 11: When the survey took place – on five days chosen from 6 months? Does it means that some respondents visited the landscape in summer and some in winter?
Response 11: Because the edible landscape of Chenggong University Town is mostly appreciated from July to December, for example, the rose garden blooms in July, the Baozhu pear is in season from September to October, and the ginkgo blossoms in December. In addition, students take winter vacation in January of their second year, so we surveyed participants on 5 days over 6 months so as to include most edible landscape categories in the experience survey.
We have added an explanation of this in the third paragraph of section 3.2: “Because of the different blooming seasons for the different types of campus edible landscapes in Chenggong University Town, the A five-day formal questionnaire survey was conducted in Chenggong University Town on five days from July to December 2021 (July 20, August 25, September 27, October 20, and December 5). A total of 460 questionnaires were distributed, and all of them were collected, excluding six invalid questionnaires with incomplete answers. Finally, A total of 454 valid questionnaires were obtained, with an effective rate of 98.7%.”
Point 12: How were the respondents recruited – were they volunteers who have answered some advert or have they been asked personally?
Response 12: Respondents were selected through random visits and sampling in various campuses in Chenggong University Town, and they filled out the questionnaire through one-to-one inquiries. Based on your suggestion, we have added information to the end of the first paragraph of section 3.2.
Point 13: 215-226 This section belongs rather to ‘Introduction/state of research’.
Response 13: Based on your suggestion, we have moved this section to “2.3. Structural equation modeling method” in “2. Literature review.”
Point 14: 227-238 This section belongs rather to ‘Aim’.
Response 14: Thank you very much for your comment. Based on your suggestion, we have moved the text on the improvement of the SEM method to the last paragraph of the introduction. We have only retained the steps of how to improve the SEM method in section 3.3.
Point 15: 227-238 This section belongs rather to ‘Aim’.
Response 15: Thank you very much for your comment. Based on your suggestion, we moved this section to the end of “2.1 Campus edible landscape” in “2. Literature review.”
Point 16: Figure 4. The part describing which ‘basic condition’ from the ‘surface value’ influences which perception type may be interesting, but unfortunately is not highlighted in the results and conclusions.
Response 16: Based on your suggestion, we have added the following passage to the end of the third paragraph of the conclusions section.
“Surface values, such as recreation, ecology and sustainability, economic benefit, and research and education, are the basic conditions for generating emotional identification.”
In addition, we have also made some additions to the conclusion section.
“Combined with the function and value of campus edible landscapes, students can participate in the process of planning, planting, cultivating, and picking of edible plants, and they can help to make food from them. Students can also put forward their own suggestions and ideas for the construction of campus edible landscapes. In the whole process, students gain a sense of happiness, which enhances their sense of belonging and attachment to the university. Thus, it is helpful to cultivate students’ practical ability and though training; this can create a good emotional effect, and build the characteristic college brand.”

Round 2
Reviewer 2 Report
Concerning the response 2
The authors wrote that the case of Chenggong University Town can provide the reference for other university towns to plan edible landscapes, so maybe it would be better to restructure the article to make it sound more like the guidelines to plan ideal university edible landscape on the base of the Chenggong University experience.
Concerning the response 3
How could the authors have improved questionnaire after the study has been completed? It is still unclear what they mean by ‘modified commonly used standard questionnaire’.
The conclusions are still unclear; the results do not support them.
The paper lacks highlighting: how the results fill the potential knowledge gap in the field of edible landscape.
Author Response
Response to Reviewer 2 Comments
Dear reviewer,
We appreciate the time and effort you have put in to reviewing our manuscript (number: ijerph-1846139). We thank you for your detailed comments, which have helped us improve the quality of this paper. In the following text, we address your concerns in the order they were provided.
Sincerely,
Qi Yi
Point 1: Concerning the response 2
The authors wrote that the case of Chenggong University Town can provide the reference for other university towns to plan edible landscapes, so maybe it would be better to restructure the article to make it sound more like the guidelines to plan ideal university edible landscape on the base of the Chenggong University experience.
Response 1: Your comments are very thoughtful and helpful. Rather than restructure the article, we have revised the relevant statement in the revised article and deleted the statement that Chenggong University Town is a typical case of campus edible landscape planning.
Point 2: Concerning the response 3
How could the authors have improved questionnaire after the study has been completed? It is still unclear what they mean by ‘modified commonly used standard questionnaire’.
Response 2: We have supplemented the detailed design process of the questionnaire in the first paragraph of Section 3.2.
“The questionnaire was based on the classic questionnaire about ‘perception-identification-behavioral intention’ commonly used by most Chinese scholars; we selected three latent variables of perception, emotional identification, and behavioral intention in this article. In the latent variable of perception, we added observed variables of functional perception and beauty perception. In the latent variable of emotional identification, we added an observed variable of likeness. In the latent variable of behavioral intention, we added an observed variable of consumer behavior. Then, we designed specific item contents around the research objectives. We finalized the questionnaire based on the above.”
We hope this addresses your concern.
Point 3: The conclusions are still unclear; the results do not support them.
Response 3: Based on your suggestions, we have logically sorted out the conclusions and made corresponding additions and corrections.
In the first paragraph of the Conclusion, we primarily summarize and explain the changes of the variables in the process of model modification and verification.
In the second paragraph, we present a comparative analysis of the hypothetical paths in the Results.
In the third paragraph, we primarily summarize the association mechanism.
In the fourth paragraph, we present the research highlights and outlook.
In the fifth paragraph of the Conclusion, we offer suggestions and reflections from the perspectives of universities and teachers and students from a practical application perspective.
Point 4: The paper lacks highlighting: how the results fill the potential knowledge gap in the field of edible landscape.
Response 4: Based on your suggestions, we have added highlights and the research outlook in the last paragraph.
Looking forward to the research progress of campus edible landscape and emotional geography, domestic and foreign research on edible landscapes has focused primarily on planning and design. The attempt to explore the emotional identification mechanisms of campus edible landscapes from the perspective of emotional geography is relatively new. From the perspective of university towns, this study enriched the research content of edible landscapes by exploring the emotional identification mechanisms of campus edible landscapes. At the same time, this study was also an exploration of the research paradigm regarding emotional geography with the hope of increasing the importance of human emotion and landscape research.
Thank you very much for your comments. Your review comments have been of great help in improving and revising this article. Based on your suggestions, we have added valuable discussion to the article. We hope our revision makes the article clearer so that more people can appreciate it.
